# Zinc, Copper, and Iron in Selected Skin Diseases

**DOI:** 10.3390/ijms25073823

**Published:** 2024-03-29

**Authors:** Aleksandra Podgórska, Aleksandra Kicman, Sylwia Naliwajko, Marta Wacewicz-Muczyńska, Marek Niczyporuk

**Affiliations:** 1Department of Aesthetic Medicine, Medical University of Bialystok, 15-267 Bialystok, Poland; aleksandra.podgorska@sd.umb.edu.pl (A.P.); olakicman@gmail.com (A.K.); marek.niczyporuk@umb.edu.pl (M.N.); 2Department of Bromatology, Medical University of Bialystok, 15-222 Bialystok, Poland; sylwia.naliwajko@umb.edu.pl; 3Department of Specialist Cosmetology, Medical University of Bialystok, 15-267 Bialystok, Poland

**Keywords:** zinc, copper, iron, trace elements, dermatosis, skin diseases

## Abstract

Trace elements are essential for maintaining the body’s homeostasis, and their special role has been demonstrated in skin physiology. Among the most important trace elements are zinc, copper, and iron. A deficiency or excess of trace elements can be associated with an increased risk of skin diseases, so increasing their supplementation or limiting intake can be helpful in dermatological treatment. In addition, determinations of their levels in various types of biological material can be useful as additional tests in dermatological treatment. This paper describes the role of these elements in skin physiology and summarizes data on zinc, copper, and iron in the course of selected, following skin diseases: *psoriasis*, *pemphigus vulgaris*, *atopic dermatitis*, *acne vulgaris* and *seborrheic dermatitis*. In addition, this work identifies the potential of trace elements as auxiliary tests in dermatology. According to preliminary studies, abnormal levels of zinc, copper, and iron are observed in many skin diseases and their determinations in serum or hair can be used as auxiliary and prognostic tests in the course of various dermatoses. However, since data for some conditions are conflicting, clearly defining the potential of trace elements as auxiliary tests or elements requiring restriction/supplement requires further research.

## 1. Introduction

Trace elements are elements found in the human body in microscopic amounts, and they are essential to the functioning of the body mainly in connection with enzymatic activity. There are 70 different trace elements in humans, and it is estimated that they account for 0.005–0.01% of body weight [1,2]. Of the 70 trace elements, 20 of them are essential for human life, with a special role assigned especially to iron, copper, and zinc [3,4,5,6]. Trace element concentrations are physiologically maintained in equilibrium. Deficiencies of these elements are associated with the development of a number of diseases such as anemia, cardiovascular or nervous system diseases, and endocrine disorders [3,4,7,8]. On the other hand, however, if the intake of some of trace elements exceeds the so-called critical level, they begin to have a toxic effect on the functioning of the human body and also affect the development of numerous diseases [2,3,8,9].

Trace elements are essential for normal skin function [10,11,12,13]. These elements have been shown to be involved in transcription of growth factors, cytokines, collagen synthesis, maintaining an oxidation–reduction balance, and also mediate wound healing [11,12,13,14]. Although the role of trace elements in skin physiology has been demonstrated, their role in the onset of skin diseases is unclear. In the course of various skin diseases, changes are observed in the concentrations of trace elements in different types of biological material, which may indicate their involvement in the pathogenesis of these conditions [15,16,17]. Appropriate supplementation or the restriction of intake of the relevant elements may have an impact on the treatment of particular skin diseases, and changes in the concentrations of these compounds may have potential as an auxiliary test in dermatology.

The aim of the present study was to gather current knowledge on changes in the directions of concentrations of the most important trace elements, zinc, copper, and iron, in the course of the most common skin diseases and to determine their potential as an additional diagnostic test. In addition, this paper outlines the precise role of zinc, copper, and iron in skin physiology (Figure 1).

## 2. Zinc

### 2.1. Biological Roles of Zinc in the Human Body

Zinc, right after iron, is the second most important trace element and is essential in maintaining the body’s homeostasis. More than 300 zinc-containing proteins have been identified in the human body, which translates into the participation of this element in many physiological processes, an example of which is erythrocyte carbonic anhydrase which requires the presence of zinc [18,19,20,21,22]. Zinc is involved in cell cycle regulation, DNA replication, repair and translation, maintenance of normal chromatin structure, and also mediates cellular processes such as proliferation, differentiation, and apoptosis. The element has also been shown to play an important role in lipid and carbohydrate metabolism and is essential for wound healing. Importantly, zinc regulates the functioning of the immune system mainly by regulating the maturation of T and B lymphocytes, natural killer cells, and dendritic cells, and it is also involved in antibody production, antigen presentation, and the process of phagocytosis. In addition, this element has antioxidant and anti-inflammatory properties [18,19,20,21,22,23].

The adult body contains 2 to 3 g of zinc, with 85% stored in muscle and bones, 11% in skin and liver, and the remainder in other tissues [18]. The human body strictly regulates the absorption and excretion of zinc. When dietary zinc is limited, zinc absorption in the duodenum and jejunum rises to 90%, while when the amount of zinc in the diet is high, its excretion is facilitated. Zinc is eliminated in small amounts in the urine (about 0.5 mg/day). In patients with liver cirrhosis or diabetes, this amount increases. Zinc excretion also occurs in feces and sweat [24,25,26]. Foods with the highest zinc content include red meat, oysters, shellfish, and whole-grain products [18,19,20,21,23].

### 2.2. Zinc in Skin Physiology

Within the skin, a greater amount of zinc is found in the epidermis (60 μg/g of tissue) compared to the dermis (40 μg/g of tissue). In the case of the dermis, the highest amounts of the element are found in the upper layers, due to the accumulation of zinc-rich mast cells [27,28]. Some of the most important biological actions of zinc include epithelial regeneration. Under the influence of skin damage, there is a release of extracellular Zn2+ ions which activate G protein-coupled receptor 39 (GPR39) and zinc-sensing receptor (ZnR), which are involved in the repair of damaged epithelium [10,27,29]. The role of this element in skin physiology is also related to the proliferation, differentiation, and survival of keratinocytes and the modulation of the release of biologically active compounds by these cells (including ATP, pro-inflammatory cytokines, and nitric oxide) [10,27,28]. Zinc’s involvement in the formation and maintenance of the proper functioning of the dermal–epidermal barrier [28], in the modulation of the immune and antioxidant response in various layers of the skin [10,11,29], and collagen synthesis [11,30] is also confirmed. Zinc also affects the activity of enzymes such as 1 and 2 5α-reductase, through which it has an anti-androgenic effect [30], and is also involved in wound healing [11,31,32].

### 2.3. Zinc in Psoriasis and Pemphigus Vulgaris

Psoriasis is a chronic, non-infectious autoimmune skin disease. It is estimated that approximately 125 million people (2–3% of the population) worldwide have psoriasis [33,34]. Clinically, the skin shows regularly spaced, symmetrical plaques covered with silvery scales, located most often on the knees, elbows, scalp, and sacral region. Psoriasis mainly involves the skin and, to a lesser extent, the joints; however, inflammation initiated in the skin can affect other organs, thus affecting the initiation and progression of other conditions. Therefore, psoriasis is considered a systemic disease. Psoriasis contributes to a dramatic reduction in patients’ quality of life and, combined with an increased risk of systemic diseases such as coronary heart disease, hypertension, or diabetes mellitus, can lead to premature death [35,36,37].

The vast majority of studies related to zinc levels in patients with psoriasis have been conducted on serum. This material is readily available using minimally invasive methods (elbow vein sampling). Serum zinc was determined using the atomic absorption spectrometry method [38,39,40,41,42,43,44,45,46,47,48] or colorimetric method [40,45]. According to most studies, patients with psoriasis have lower serum concentrations of this element [16,38,39,40,41,42,43,44,45,46,47]. In addition, patients with involvement of more than 20% of the body surface and a positive history of psoriasis had statistically significantly lower serum zinc concentrations [48]. Concentrations of this element also correlated with disease severity according to the PASI classification, which may indicate the potential of zinc as an adjunctive test in the diagnosis and classification of psoriasis [17]. Zinc concentrations also vary between psoriasis types, with the lowest zinc levels found in palmoplantar psoriasis [42]. Low zinc concentrations affect the levels of zinc transporters ZIP2 and ZIP3, and patients with psoriasis show an increase in blood levels of these proteins, likely in response to low zinc concentrations [49].

On the other hand, some reports indicate that, between healthy individuals and psoriasis patients, there are no differences in zinc concentrations [17,48,50,51,52]. In these studies, zinc was determined using atomic absorption spectrometry [17,48,49] or with the colorimetric method [50]. There was also no correlation between zinc concentrations and clinical features of the disease [48]. These differences may be due to differences in the methodology of zinc determination and the selection of study groups (differences in ethnicity and diet). Nevertheless, preliminary studies point to a role for zinc in the pathogenesis of psoriasis and indicate that normalizing levels of this micronutrient may have a beneficial effect in treating patients with the disease.

Another autoimmune skin disease is pemphigus. This group of diseases is characterized by intraepithelial clefting and acantholysis. Some of the most common forms of pemphigus include pemphigus vulgaris (PV) [53]. Patients most often report the presence of blisters and erosions that involve the skin and mucous membranes (mainly in the mouth). Blisters on the skin are resistant to treatment and develop into oozing erosions [53,54,55].

As in previous studies, zinc was determined colorimetrically [56] or with atomic absorption spectrometry [57,58,59]. The number of reports on serum zinc concentrations in PV patients is sparse. However, lower zinc concentrations are observed in patients with the disease, which may indicate that low concentrations of this element may be involved in the pathogenesis of PV [56,57,58,59]. Unfortunately, no correlation was found between zinc concentrations and disease duration and activity [59]. However, it should be emphasized that this is a single study, and further research on the role of zinc in PV and its potential in determining the duration and severity of the disease should be considered in the future.

### 2.4. Zinc in Atopic Dermatitis

Atopic dermatitis (AD) is one of the more common inflammatory skin conditions. The disease is characterized by periods of exacerbation and remission. The skin manifestations of AD include erythematous, scaly, and itchy lesions that are most often located on the volar surfaces of the extremities [60,61]. AD mainly affects children, and the incidence in the world population is steadily increasing, which is mainly related to the multifactorial pathogenesis of the condition [62]. As with psoriasis, the disease is associated with a deterioration in quality of life [60,61].

Among AD patients, zinc has been determined in different types of material, including serum, erythrocytes, or hair. In patients with atopic dermatitis, serum zinc was determined using atomic absorption spectrometry [63,64,65,66,67,68] or the proton-induced X-ray fluorescence technique in the case of hair [64]. Unfortunately, data on zinc in the course of this disease are inconsistent. According to some studies, there are no differences in serum zinc concentrations between AD patients and healthy individuals [63,64,65,66,67,68]. A single study indicates that serum zinc concentrations do not depend on the severity of AD [67]. On the other hand, however, many researchers indicate that patients with AD are found to have lower serum concentrations of this element [69,70,71,72,73,74]. Again, a single study shows a correlation between serum zinc levels and AD severity; patients with more advanced forms of the disease had lower serum levels of this element compared to those with milder forms of the disease [74,75].

More consistent are the reports related to zinc determined in erythrocytes; patients with AD had lower zinc concentrations compared to healthy subjects, and, in addition, a negative correlation was shown between levels of this element and the scoring index for atopic dermatitis (SCORAD) [68,70]. Importantly, studies of the hair of AD patients have also shown lower levels of zinc [64,69,76]. This indicates that zinc may be involved in the pathogenesis of AD and may be used in the future as an auxiliary test in the diagnosis and evaluation of this disease; however, the target material for determination must be full blood or hair.

### 2.5. Zinc in Acne Vulgaris

Acne (acne vulgaris) is a chronic inflammatory disease of the hair follicles, affecting more than 9.4% of the world’s population. Acne affects more than 85% of adolescents but can also appear in adults [77]. Acne lesions can be divided into non-inflammatory (blackheads) or inflammatory (papules, pustules, or cysts), and the treatment of acne requires long-term and sometimes persistent therapy [77,78].

In the case of acne, the direction of changes in zinc concentrations appears to be unidirectional. Zinc was also determined using atomic absorption spectrometry in patients with acne [79,80,81,82,83,84,85,86,87,88,89,90,91]. The available literature data indicate that patients with acne have lower serum zinc concentrations [79,80,81,82,83,84,85,86,87,88,89,90,91]. According to Butool et al. [80], zinc deficiency may be one of the main causes of acne. The use of zinc may be an auxiliary method for treating this disease [92].

Although patients with acne have lower serum zinc concentrations, reports on the correlation of this element with disease severity are conflicting. According to Saleh et al. [81], Rostami Mogaddam et al. [87], Ozuguz et al. [91], and Thomas [93], zinc concentrations correlated with acne severity; patients with more severe forms of the disease had lower concentrations of this element. On the other hand, however, according to studies by Kaymak et al. [84] and Goodarzi et al. [85], zinc concentrations did not correlate with disease severity. It should be noted, however, that in the study by Goodarzi et al. [82], the lack of the described relationship was found only in men. Also, no correlation was found between zinc concentrations and acne duration [84,87]. However, one study indicated that there is a correlation between serum zinc and the type of acne lesions such as blackheads on the left cheek, papules on the forehead, chest and upper back, and pustules on the right cheek, chin, chest, and upper back [87].

### 2.6. Zinc in Seborrheic Dermatitis

Another inflammatory skin disease is seborrheic dermatitis (SD), which mainly affects skin areas rich in sebaceous glands. SD occurs in both adults and children, as well as in infants, and differs in the clinical manifestations in each group. In adults and children, symptoms range from mild patches to diffuse desquamation, while in infants SD manifests as scaly yellow patches [94].

It is unfortunate that the number of reports on the serum zinc concentrations of SD patients is severely limited. As reported by Aktaş et al. [95], patients with SD have lower serum zinc concentrations (atomic absorption spectrometry method); however, observations by Zohreh et al. [96] and Nazik et al. [15] indicate that there are no differences in zinc concentrations between sick and healthy patients (atomic absorption spectrometry was used in both studies). In addition, a single study indicates higher zinc concentrations in the hair of SD patients [15].

## 3. Copper

### 3.1. Biological Roles of Copper in the Human Body

Copper is a trace element essential for maintaining organism homeostasis. This element functions as a cofactor or building component of numerous enzymes involved in biological processes, mainly related to energy metabolism (e.g., cytochrome c oxidase), the oxidation–reduction system (e.g., superoxide dismutase) and iron turnover (hephaestin and ceruloplasmin) [97,98,99,100]. Copper is essential in biological processes such as the synthesis of hemoglobin, neurotransmitters, myelin, melanin, aerobic respiration, thyroid metabolism, normalizing calcium, and phosphorus concentrations and the formation of connective tissue [98,101,102]. This element has both pro- and antioxidant effects, depending on biological conditions. As an antioxidant compound, it participates in neutralizing free radicals; however, excessive copper activity promotes the formation of free radicals, thereby contributing to cell and tissue damage [102,103]. In addition, copper exhibits strong antimicrobial activity, thus acting as a compound with protective properties. Among others, copper eliminates *E. coli* O157:H7, methicillin-resistant *Staphylococcus aureus* (MRSA), *Clostridium difficile*, influenza A virus, adenoviruses, and fungi [104].

The human body contains between 50 and 120 mg of copper, and most of this element is found in muscle, bone, and liver [100,101]. Humans supply copper to the body through food and water, and it is absorbed in the small intestine (contribution of high-affinity Cu transporter 1 [CTR1]) and then transported to the liver, which is responsible for the distribution and homeostasis of this element. In the bloodstream, it is transported with the help of ceruloplasmin and albumin. The main route of copper elimination occurs through the biliary tract. The element is released into the bile, where it then enters the intestines and finally into the feces. The amount of copper released into the bile depends on an individual’s characteristics. It is estimated that, with feces, approximately 0.6–1.6 mg of copper is removed. A small amount of this element is also removed in the urine (0.05 mg) [105,106]. Foods rich in this element include beef liver, nuts (especially cashews), fish (mackerel, cod), and turkey and chicken meat [99,100,101].

### 3.2. Copper in Skin Physiology

Copper also has a number of important biological functions within the skin. Some of this is due to the antioxidant potential of this element, as mentioned earlier. Acting as a cofactor of superoxide dismutase, it protects the skin from free radicals, mainly from cell membrane damage and lipid peroxidation [12,107]. The element is also a cofactor of lipoxygenase, thus enabling proper cross-linking of extracellular matrix proteins and their stabilization [12], as well as tyrosinase, an enzyme responsible for the biosynthesis of the pigment melanin, which is essential for normal skin pigmentation [12,108]. Copper stimulates the proliferation of fibroblasts which are among the most important building and secretory cells in the skin [12,109,110] and increases the production of type I, II, V collagen, elastin, and fibrillin by these cells [12,107,111]. Importantly, copper has been shown to affect the production of transforming growth factor β (TGF-β) by fibroblasts [12,109,111], a protein that plays an important role in maintaining skin homeostasis, including by inhibiting keratinocyte proliferation and regulating cell differentiation [112]. Another biologically active molecule whose activity in the skin depends on copper is the collagen-related heat shock protein HSP47, a protein essential for collagen synthesis [12,111,113]. As mentioned earlier, copper’s antibacterial properties have a protective effect on the skin [12,114], and a role for copper in wound healing has also been demonstrated [114,115].

### 3.3. Copper in Psoriasis and Pemphigus Vulgaris

Copper determined from serum may be a potential diagnostic and prognostic marker in patients with psoriasis. According to most available studies, patients with psoriasis have higher serum copper levels compared to healthy individuals (copper was determined using atomic absorption spectrometry) [16,38,39,43,48,50,51,116,117,118,119,120]. However, a single study indicates that there are no differences between the serum copper concentrations of psoriasis patients and healthy individuals [117]. Reports related to correlations between copper concentrations and disease severity are a matter of debate. According to Aggarwal et al. [50], Shahidi-Dadras et al. [117], and Rashmi et al. [118], patients with more advanced forms of the disease have higher serum copper levels compared to those with less advanced forms of the disease. In addition, copper concentrations correlated positively with the psoriasis area and severity index (PASI), which assesses the severity of the disease. The described correlation was also confirmed using Pearson correlation analysis [50,117]. A two-study conducted by Mohammad et al. [17] and Khan et al. [45] does not indicate a correlation between copper concentrations and psoriasis severity. The vast majority of studies indicate that patients with psoriasis have higher serum concentrations of this element, which may indicate the potential of copper in the diagnosis or potential treatment of the disease. It is unfortunate that we do not have data on urinary copper concentrations in patients with psoriasis. However, assessment of copper in urine (in daily urine collections) is used, for example, in the diagnosis of Wilson’s disease; this indicates the potential in determining this element also in patients with psoriasis [121]. Other postulated biological materials that may be used in the future to determine copper in psoriasis patients include hair. Preliminary reports on the feasibility of determining copper in hair in dermatological patients have been demonstrated from studies of patients with atopic dermatitis [69]. This will be presented in the following sections of the paper.

According to some studies, there are no differences in serum copper concentrations between patients with PV and healthy subjects [56,58,59,122]. A single study also looked at patients with pemphigus foliaceus; however, again, no differences in copper concentrations were found between the two groups of patients [59]. It should be noted, however, that some studies indicate that patients with PV have lower serum copper concentrations compared to healthy individuals [57,123]. In addition, one study found a negative and significant correlation between copper concentrations and disease duration in men [56]. Differences in copper concentrations may be due to differences in the ethnicity of patients, unequivocally establishing a relationship between serum copper concentrations and the presence of PV and the severity of the disease.

### 3.4. Copper in Atopic Dermatitis

The number of reports on copper concentrations in AD patients is limited and contradictory. As reported by el-Kholy et al. [69], David et at. [63], and Al-Ghurabi et al. [124], higher serum copper concentrations are found in patients with atopic dermatitis compared to healthy patients. One study indicates that higher copper concentrations are also observed in the hair of patients with AD [69]. On the other hand, according to studies by Toyran et al. [70], Zackheim et al. [116], and Toro et al. [64], there are no differences in serum concentrations of this element between healthy and diseased individuals. Moreover, a single study by Hon et al. [71] indicates that 40% of patients with AD have low serum copper levels. All studies used atomic absorption spectrometry to determine copper concentrations. Unambiguous determination of copper’s potential as an auxiliary test in the diagnosis of patients with AD requires further testing.

### 3.5. Copper in Acne Vulgaris

As with AD, data on serum copper concentrations in acne patients are conflicting (copper determined by using atomic absorption spectrometry only). On the one hand, most reports indicate that acne patients have lower serum copper concentrations compared to healthy individuals [80,82,125,126,127,128,129]. In addition, according to the study by Khayyat et al. [127], the concentration of this element decreased as the severity of acne increased. On the other hand, some studies have reported that there are no differences in serum copper concentrations between acne patients and healthy individuals [81,83]. It should be noted, however, that according to most studies, lower copper concentrations are observed in patients with acne, which indicates the role of this element in the pathogenesis of acne.

### 3.6. Copper in Seborrheic Dermatitis

Currently, there are few reports on copper levels in SD patient samples (copper determined by using atomic absorption spectrometry only). However, all studies indicate that patients with SD have higher serum copper concentrations compared to healthy individuals [15,96,130,131]. Interestingly, in patients with SD, a negative correlation was found between serum copper and calcium levels. This relationship was not found in healthy subjects. This may indicate that copper and calcium concentrations are interrelated in the pathogenesis of this disease [130]. Importantly, high copper concentrations are also found in hair samples of SD patients, indicating that the determination of this element in hair may have potential as an adjunctive test in the diagnosis of SD [15].

## 4. Iron

### 4.1. Biological Roles of Iron in the Human Body

Iron, unlike zinc, is an element abundant on earth and biologically essential for living organisms. Its concentration in tissues, due to its ability to form free radicals, must be strictly regulated to prevent tissue damage resulting from excessive amounts of it [132,133]. The regulation of iron in the body is maintained by the amount of iron absorbed rather than by its elimination. It does not have an active physiological excretion process. A peptide produced in the liver—hepcidin—helps regulate iron levels. High concentrations of the element stimulate the production of the peptide, which reduces iron absorption and prevents iron overload in the body, and, conversely, low levels of it reduces the production of hepcidin, increasing the absorption of iron from the diet and releasing the element stored in the body into the circulation [134,135]. Iron plays an important role in the human body. It is an essential component of hemoglobin, which binds and transports oxygen from the lungs to the entire body. It is also a component of myoglobin, a muscle protein that ensures adequate oxygen supply to muscles. In addition, it is bound in enzymes involved in the oxidation–reduction reaction or electron transport chain, as well as cytochromes and catalase. Iron is also a cofactor in the synthesis of DNA, amino acids, and hormones. Iron–sulfur complexes, being part of aconitase, NADH dehydrogenase, and succinate dehydrogenase, are involved in energy production in mitochondria. Iron also has an impact on the body’s immune function [136,137,138].

Daily iron requirements depend on a person’s age, gender, and physiological state, among other factors [139]. The recommended daily allowance (RDA) for iron ranges from 0.27 mg in infancy to 27 mg during pregnancy. Menstruating women should take 18 mg of iron per day, while women during childbearing years and adult men should take 8 mg [140]. Iron is absorbed in the small intestine, primarily in the duodenum and jejunum. It is not synthesized in the body and, hence, must be supplied either with food or in the form of supplements. Non-heme iron is found in cereals, legumes, fruits, vegetables, nuts, and chocolate, while heme iron is found in meat and seafood. Heme iron shows better absorption and bioavailability than non-heme iron [136].

### 4.2. Iron in Skin Physiology

Iron is an essential element for healthy skin, mucous membranes, hair, and nails. Its deficiencies can manifest as pale skin, itching, susceptibility to skin infections, inflammation of the corners of the mouth, and brittle nails and hair, among other symptoms [141]. Skin is involved in the storage and removal of iron from the body [142]. Its concentration in healthy human skin reaches about 0.15–0.275 mg/g [143]. The content of iron in the epidermis varies in different layers of the epidermis and depends on the degree of differentiation of keratinocytes. The highest content was found in the basal layer and the lowest in the stratum corneum. In the dermis, iron levels also vary and increase with skin aging [13,144]. Iron is excreted from the epidermis through the stratum corneum in the process of epidermal exfoliation. The element enters the outer layer of the epidermis; the stratum corneum is a component of keratinocytes, which undergoes keratinization and, thus, leaves the lower granular layer or pass through the intercellular space [145]. Iron, as a transition metal, is involved in the process of oxidative stress in the skin caused by reactive oxygen species (ROS), which are formed in the skin when exposed to UVA radiation. Exposure of fibroblasts present in the skin to this radiation generates ROS, which cause oxidative damage in mitochondrial, lysosomal, plasmatic, and nuclear membranes. The consequence is loss of cell membrane integrity and mitochondrial-derived energy (ATP), further leading to necrotic skin cell death [146,147]. Iron also affects hair color. It has been shown that people with red and dark brown hair have higher iron levels. In addition, excess iron in the tissues and abnormalities in its metabolism results in hair loss. Deficiency, on the other hand, results in a temporary loss of hair color due to a lack of melanin in the hair shaft. It has also been shown that iron contamination of the workplace can cause darkening of the face and nails [148,149,150].

### 4.3. Iron in Psoriasis and Pemphigus Vulgaris

The number of studies on the role of iron in psoriasis is limited; however, they point to an important role for this element in the course of psoriasis. Studies indicate that patients with psoriasis of mild to severe severity have lower iron content in the epidermis compared to healthy individuals (again determined using the absorption spectrometry). However, in the dermis, its content was higher in psoriasis patients compared to healthy subjects. The decrease in epidermal iron content was associated with excessive proliferation and epidermal peeling in psoriatic patients [151,152].

At the same time, studies indicate that lower iron concentrations are also observed in the serum of psoriasis patients [117,153]. The role of low iron concentrations in psoriasis is indicated by changes in other iron-related laboratory parameters such as a decrease in the transferrin iron saturation index, hepecidin, and an increase in the soluble receptor, which for transferrin indicates a negative iron balance in the body [154].

Currently, there are no scientific reports on the role of iron in pemphigus. However, identifying potential changes in concentrations of this element may have diagnostic potential in patients with psoriasis. However, this requires research.

### 4.4. Iron in Atopic Dermatitis

Studies have shown that a predisposition to iron deficiency can be inherited from mother to child, so levels of this element are extremely important in pregnant women and determine the later risk of atopy in children. Normal iron levels in expectant mothers reduce the risk of AD in children [155,156,157,158]. Iron deficiencies affect the immune response, stimulating immune cells to become overactive [159].

Despite the fact that iron deficiencies are associated with the risk of developing AD, no changes in the concentration of this element are found in patients with the disease compared to healthy individuals [63,70,160]. In these studies, iron was determined using atomic absorption spectrometry [60,67] or inductively coupled plasma (mass spectrometry) [160]. Additionally, iron concentrations did not correlate with the severity of AD (atomic absorption spectrometry method) [75].

### 4.5. Iron in Acne Vulgaris

In contrast to copper and zinc, the amount of data on iron concentrations in patients with AV is very sparse. Two studies indicate that there are no differences in iron concentrations between patients with and without acne (iron determined using atomic absorption spectrometry) [126,161]. Interestingly, lower iron concentrations are observed in patients with severe forms of nodular acne, which are most likely due to the presence of chronic dermatitis. It is possible that a decrease in iron concentrations as a result of skin inflammation is likely to be observed in AV, especially severe forms, but this requires further studies.

### 4.6. Iron in Seborrheic Dermatitis

Serum iron levels in people with seborrheic dermatitis were significantly higher than in healthy subjects. Its content in hair, on the other hand, was lower in diseased subjects compared to those without seborrheic dermatitis [15,130,162]. Iron studies were performed using atomic absorption spectrometry [15,130] or the colorimetric method [162].

## 5. Further Research Directions: Medical Ultrasonography of Skin

This paper demonstrates that the exact role of trace elements in skin diseases is inconclusive; however, a significant part of research points to changes in their concentrations in skin diseases. Among the methods that could be used in the future to determine the role of these elements in skin diseases is ultrasonography (USG) of the skin. In dermatology, skin USG is one of the oldest visualizing techniques that, since it is fast and quite cheap, also is increasing in popularity among patients. Other reasons for its approval are that it is both noninvasive as well as painless. Some other advantages of skin USG are related to the utilization of non-ionizing media and a lack of contraindications to its performance [163,164]. Currently, ultrasound is widely used in the diagnosis of skin diseases such as AD or psoriasis. Various skin diseases are characterized by changes in ultrasound images [165,166].

By using ultrasonography of the skin, it will be possible to assess the condition of the skin in patients with particular skin diseases and to establish the correlation of skin conditions with the concentrations of trace elements determined in different types of biological materials. The additional introduction of an appropriate diet in dermatological patients, such as a diet rich or poor in a given trace element or the use of supplementation and the subsequent determination of changes in the state of the skin by ultrasound, will also allow us to expand our knowledge of the role of these elements in skin diseases and may also be an additional method in supporting primary treatment.

## 6. Conclusions

Elements such as zinc, copper, and iron are essential in the functioning of the human body. They have also been shown to be essential for the proper functioning of the skin and its components. However, the role of trace elements in the pathogenesis of dermatological diseases is still not fully explored. Some studies indicate the role of zinc, copper, and iron in the course of various dermatoses, which may indicate the potential for supplementation or the restriction of intake of these compounds as part of the treatment of these conditions and the possibility of using trace element determinations as an adjunctive test in dermatology. However, reports on this subject are conflicting, as shown in Table 1. Although a large number of studies indicate that normal concentrations of zinc, copper, and iron are disrupted in the course of skin diseases and that the determinations of these elements can be used as auxiliary tests in dermatology, a clear determination of their potential requires further research.

## Figures and Tables

**Figure 1 ijms-25-03823-f001:**
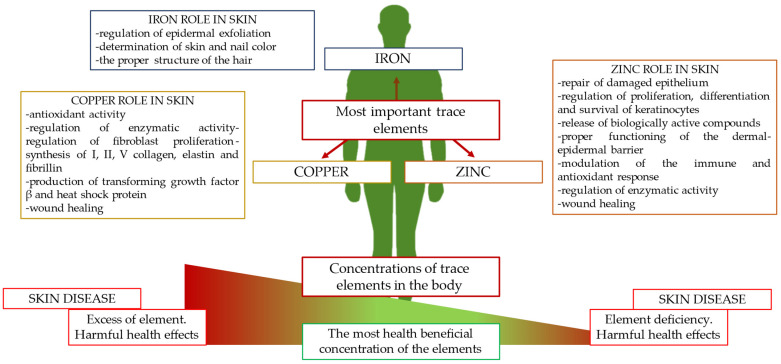
The most important trace elements in human physiology and their functions in skin.

**Table 1 ijms-25-03823-t001:** Role of zinc, copper, and iron in skin diseases.

Psoriasis	
ZINC	
Material	Observations
**Serum**	Conflicting reports - Lower zinc levels in patients with psoriasis [16,38,39,40,41,42,43,44,45,46,47] or - No differences in zinc levels between psoriasis patients and healthy patients [17,48,49,50,51,52] - Lower zinc concentrations in patients with more than 20% skin involvement and a positive family history [51] - Zinc concentrations correlated with disease severity and psoriasis type [17,42] or - No correlation between zinc concentrations and the course of psoriasis [48]
**COPPER**	
**Serum**	Conflicting reports - Higher copper levels in patients with psoriasis [16,17,38,39,43,48,50,51,52,116,117,118,119,120,121] or - No differences in copper levels between psoriasis patients and healthy patients [117] - Higher copper levels in patients with more advanced forms of the disease [50,117,118] or - No correlation between zinc concentrations and the course of psoriasis [17,48]
**Urine or hair**	No studies available, but this type of material can be used in the determination of copper concentrations in patients with psoriasis
**IRON**	
**Skin**	- Lower iron concentrations in the epidermis and higher in the dermis of psoriasis patients [151,152]
**Serum**	Reports sparse but consistent - Lower iron levels in patients with psoriasis [117,153]
**Pemphigus vulgaris**	
**ZINC**	
**Serum**	Reports sparse but consistent - Lower zinc levels in patients with pemphigus [56,57,58,59] - No correlation between zinc concentrations and course of pemphigus [59]
**COPPER**	
**Serum**	Conflicting reports - Lower zinc levels in patients with pemphigus [57,123] or - No differences in zinc levels between pemphigus patients and healthy patients [56,58,59,122] - Negative correlation between copper concentrations and pemphigus duration in men [56]
**IRON**	
No literature reports	
**Atopic Dermatitis**	
**ZINC**	
**Serum**	Conflicting reports - Lower zinc levels in patients with AD [69,70,71,72,73,74] or - No differences in zinc levels between AD patients and healthy patients [63,64,65,66,67,68] - Lower zinc levels in patients with more advanced AD [74,75] or - No correlation between zinc concentrations and course of AD [67]
**Erythrocytes**	Reports sparse but consistent - Lower zinc concentrations in AD patients [68,70] - Negative correlation between zinc concentrations and SCORAD index [68,70]
**Hair**	Reports sparse but consistent - Lower zinc levels in AD patients [64,69,76]
**COPPER**	
**Serum**	Conflicting reports - Higher copper levels in patients with AD [63,69,124] or - Lower copper levels in patients with AD [71] or - No differences in copper levels between AD patients and healthy patients [64,70,116]
**Hair**	Single study - Higher copper levels in patients with AD [69]
**IRON**	
**Serum**	Reports sparse but consistent - No differences in copper levels between AD patients and healthy patients [63,70,160] - No correlation between zinc concentrations and the course of AD [75]
**Acne Vulgaris**	
**ZINC**	
**Serum**	Consistent reports - Lower zinc levels in patients with acne [79,80,81,82,83,84,85,86,87,88,89,90,91] Conflicting reports - Lower zinc levels in patients with more advanced acne [81,87,91] - Correlation of zinc concentration with the occurrence of individual acne lesions [87] or - No correlation between zinc concentrations and the course of acne [84,85]
**COPPER**	
**Serum**	Conflicting report - Lower zinc levels in patients with acne [80,125,126,127,128,129] or - No differences in copper levels between acne patients and healthy patients [81,83] Single study - Correlation between copper concentrations and acne severity [127]
**IRON**	
**Serum**	Reports sparse but consistent - No differences in copper levels between acne patients and healthy patients [126,161]
**Seborrheic Dermatitis**	
**ZINC**	
**Serum**	Conflicting report - Lower zinc levels in patients with SD [95] or - No differences in copper levels between SD patients and healthy patients [15,96]
**Hair**	Single study - Higher zinc levels in patients with SD [15]
**COPPER**	
**Serum**	Reports sparse but consistent - Higher copper levels in patients with SD [15,96,130,131]
**Hair**	Single study -Higher copper levels in patients with SD [15]
**IRON**	
**Serum**	Reports sparse but consistent - Higher iron levels in patients with SD [15,130,162]
**Hair**	Single study - Higher iron levels in patients with SD [15]

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
