# Peer review of "Zinc, Copper, and Iron in Selected Skin Diseases"

_ijms, 2024, doi:10.3390/ijms25073823_

Round 1
Reviewer 1 Report
Comments and Suggestions for Authors
The analysis of trace elements in relation to skin diseases presents valuable insights but is hindered by several critical issues that require revision for clarity, credibility, and depth.
The introduction lacks a clear context or rationale for the study's importance in dermatology and fails to outline the methodology used, leaving readers uncertain about the research approach. Additionally, the study's aim is vaguely defined, lacking clear research questions or hypotheses, and the rationale for selecting zinc, copper, and iron as focus elements is not clearly explained.
Critical analysis is lacking throughout the text, with information presented without discussing potential limitations or conflicting findings. Conflicting reports on the role of trace elements are acknowledged but not analyzed, contributing to uncertainties.
Furthermore, Figure 1 does not effectively convey meaningful information and could benefit from redesign.
The sections on zinc, copper, and iron each have their own issues.
Zinc- The section focuses predominantly on serum zinc concentrations, neglecting other potential indicators or sources of zinc in the body, which may provide a more comprehensive understanding of its role in skin diseases. The section does not discuss the methodologies used in the studies cited, leaving readers unaware of potential biases or limitations in the research. Copper- Similar to the previous sections, this part lacks critical analysis of the presented data, providing information without discussing potential limitations or biases in the studies cited. The section relies heavily on serum copper concentrations as the primary indicator, neglecting other potential sources or indicators of copper levels in the body. Correlations between copper concentrations and disease severity vary across studies, leading to uncertainty about the significance of copper levels in the pathogenesis or diagnosis of skin diseases. Iron- Inconsistencies in the relationship between iron concentrations and skin diseases, such as the variation in iron levels in patients with psoriasis and the lack of significant changes in patients with atopic dermatitis or acne vulgaris, raise questions about the reliability of conclusions drawn.
The text briefly mentions the potential use of ultrasonography of the skin as a method for assessing the correlation between skin conditions and trace element concentrations. However, it does not elaborate on the feasibility, limitations, or existing research supporting this approach.
While the text outlines the potential implications of trace elements for dermatological diseases, it lacks depth in discussing specific mechanisms of action or pathways involved. Providing more detailed explanations of how zinc, copper, and iron influence skin physiology and pathogenesis would enhance the understanding of their roles in dermatology.
The conclusion rightly emphasizes the need for additional research to clarify the role of trace elements in dermatological diseases. However, it does not offer specific suggestions or directions for future studies, such as identifying key research questions, methodologies, etc.
Comments on the Quality of English LanguageModerate editing of English language required
Author Response
Reviewer 1
The analysis of trace elements in relation to skin diseases presents valuable insights but is hindered by several critical issues that require revision for clarity, credibility, and depth.
Dear Reviewer,
We would like to thank you very much for your careful review of our paper, entitled. " Zinc, copper, and iron in selected skin diseases." and for your accurate and useful suggestions. We have highlighted responses to suggestions in blue italics while changes to the manuscript have been highlighted in green. We hope that the corrections made will prove satisfactory and allow publication of our work in the “International Journal of Molecular Sciences”.
The introduction lacks a clear context or rationale for the study's importance in dermatology and fails to outline the methodology used, leaving readers uncertain about the research approach. Additionally, the study's aim is vaguely defined, lacking clear research questions or hypotheses, and the rationale for selecting zinc, copper, and iron as focus elements is not clearly explained.
Thank you for your pertinent comment. Trace elements are essential for the proper functioning of the skin, however, their unequivocal role in the pathogenesis of dermatological diseases is not clear. Our work sums up the existing knowledge on the changes of the most important trace elements (copper, zinc and iron) in different types of material. The selection of just these trace elements was not random and was supported by previous analysis of databases such as PubMed (at the stage of manuscript preparation). At the same time, we believe that the explanation for the choice of copper, iron and zinc specifically is adequate (see lines 31; 49-53) The most common test material is serum - a material that is easy to collect (peripheral blood sampling At the same time, we would like to point out that part of the work also involves hair material. Referring to the comment referring to methodology, trace elements are usually determined using atomic absorption spectrometry. However, to meet the reviewer's expectations, the name of the analytical method by which trace elements were determined in a given study will be added to the relevant sections in the manuscript. Any changes in the manuscript will be marked in green.
Critical analysis is lacking throughout the text, with information presented without discussing potential limitations or conflicting findings. Conflicting reports on the role of trace elements are acknowledged but not analyzed, contributing to uncertainties.
Thank you for your comment. The purpose of this study was to collect data on changes in trace elements in the course of selected dermatological diseases and to determine their potential as additional dermatological tests. Currently, this issue is very poorly understood and the number of scientific reports related to this topic is most often limited. The few papers most often indicate, changes in the concentration of trace elements or in the course of dermatological diseases, while not indicating the cause of these changes. We realize that most often studies indicate contradictory reports on elemental concentrations in different types of material, however, we have no influence on the obtained results. Variations in the data obtained are not explained between different papers, but may be due to many factors such as diet, race, ethnic group and many others. These are only speculations on our part, which is why we did not include them in our work.
Furthermore, Figure 1 does not effectively convey meaningful information and could benefit from redesign.
Thank you for your pertinent comment, we will redraw the figure in an appropriate way to further depict the most important roles of trace elements in skin physiology. We hope that the figure revised in this way will be more valuable to view.
Zinc- The section focuses predominantly on serum zinc concentrations, neglecting other potential indicators or sources of zinc in the body, which may provide a more comprehensive understanding of its role in skin diseases. The section does not discuss the methodologies used in the studies cited, leaving readers unaware of potential biases or limitations in the research.
Thank you for your comment, we realize that most of the studies we cited are related to serum. At the same time, we would like to emphasize that currently most studies on zinc in dermatological diseases are based on the determination of serum concentrations of this element. In preparing this manuscript, we made a thorough and factual analysis of medical databases and were unfortunate not to find any papers other than those on serum, erythrocytes or hair. The single study we were able to include concerns zinc transporters (ZIP2 and ZIP3) in patients with psoriasis. Perhaps in the future there will be studies on, for example, skin samples, however, at the moment we do not have such studies. At the same time, we believe that our work indicates the potential in using the determination of trace elements in serum as ancillary tests in dermatology. At the reviewer's request, we have supplemented the manuscript with data on the methodologies used in the determination of trace elements in different types of material. Changes are highlighted in green.
Copper- Similar to the previous sections, this part lacks critical analysis of the presented data, providing information without discussing potential limitations or biases in the studies cited. The section relies heavily on serum copper concentrations as the primary indicator, neglecting other potential sources or indicators of copper levels in the body. Correlations between copper concentrations and disease severity vary across studies, leading to uncertainty about the significance of copper levels in the pathogenesis or diagnosis of skin diseases
Thank you for your comment. Again, we would like to point out that the amount of data on the role of copper in dermatological diseases is very limited. We have included in the manuscript all the data we found related to this issue - most of the papers we used were based solely on examining changes in copper in the serum or hair of dermatological patients, in order to determine whether a particular dematosis affects the concentrations of this element. We realize that the data we collected are contradictory - depending on the work cited. However, we have no control over the data obtained and we feel that the summary table clearly and legibly sums up the scientific reports we collected in such a way as to indicate the differences obtained in the experiments and not to mislead the reader. In addition, as we wrote earlier, we are not able to say where the discrepancies in the studies come from, it may be due to the bias or limitations of the studies mentioned by the reviewer. However, changes in copper concentrations may also be related to a number of other factors such as gender, race, nationality, medications taken and many others. However, these are speculations, which is why we do not describe them in our manuscript - mainly so as not to mislead the reader by presenting unproven data.
In accordance with the reviewer's fair comment, we also pointed out in the chapter on copper the method for determining this element in the test material. The changes have been highlighted in green.
Iron- Inconsistencies in the relationship between iron concentrations and skin diseases, such as the variation in iron levels in patients with psoriasis and the lack of significant changes in patients with atopic dermatitis or acne vulgaris, raise questions about the reliability of conclusions drawn.
Thank you for your comment. It is unfortunate in describing iron concentrations we only have the work presented in the manuscript. We are not influenced by the results obtained by other teams. As we wrote in earlier sections of this response, the differences in trace element concentrations may be due to a number of factors. However, these are purely our speculations. This manuscript is intended to present the changes in trace element concentrations in the body fluids of patients with skin diseases and to point out their potential as ancillary studies. It is possible that new studies will emerge in the future that will indicate whether changes in the concentration of these elements are one of the causes of the onset of the disease or an effect of its development. We have reposted the iron determination methodology in the manuscript. The changes have been highlighted in green.
The text briefly mentions the potential use of ultrasonography of the skin as a method for assessing the correlation between skin conditions and trace element concentrations. However, it does not elaborate on the feasibility, limitations, or existing research supporting this approach.
Thank you for your pertinent comment. As we pointed out in the paper, the exact role of trace elements in skin diseases is currently unclear. The reviewer rightly pointed out that most of the studies we presented are related to serum. Performing skin ultrasound, especially in combination with the determination of elements in blood, could be a good complementary method. Skin ultrasound is a method that is increasingly being used in dermatology, however, no one has currently done a study that links skin ultrasound and the role of trace elements in dermatoses. However, our research team is now beginning experiments on the use of ultrasound and serum trace element concentrations in patients with selected skin diseases. The subsection "Further Research Directions - Medical ultrasonography of skin" provides our perspective on the possibilities arising from skin ultrasound. In accordance with the reviewer's suggestion, we will enrich the chapter with basic data on skin ultrasound and also the limitations and advantages of this method. Changes have been highlighted in green. Added literature:
- Csány, G.; Gergely, L.H.; Kiss, N.; Szalai, K.; Lőrincz, K.; Strobel, L.; Csabai, D.; Hegedüs, I.; Marosán-Vilimszky, P.; Füzesi, K.; et al. Preliminary Clinical Experience with a Novel Optical–Ultrasound Imaging Device on Various Skin Lesions. Diagnostics 2022, 12, 204, doi:10.3390/diagnostics12010204.
- Barcaui, E.D.O.; Carvalho, A.C.P.; Lopes, F.P.P.L.; Piñeiro-Maceira, J.; Barcaui, C.B. High Frequency Ultrasound with Color Doppler in Dermatology. A Bras Dermatol 2016, 91, 262–273, doi:10.1590/abd1806-4841.20164446.
- Sorokina, E.; Mikailova, D.; Krakhaleva, J.; Krinitsyna, J.; Yakubovich, A.; Sergeeva, I. Ultrasonography Patterns of Atopic Dermatitis in Children. Skin Research and Technology 2020, 26, 482–488, doi:10.1111/srt.1283
- Șomlea, M.; Boca, A.; Pop, A.; Ilieș, R.; Vesa, S.; Buzoianu, A.; Tătaru, A. High-Frequency Ultrasonography of Psoriatic Skin: A Non-Invasive Technique in the Evaluation of the Entire Skin of Patients with Psoriasis: A Pilot Study. Exp Ther Med 2019, doi:10.3892/etm.2019.8140.
While the text outlines the potential implications of trace elements for dermatological diseases, it lacks depth in discussing specific mechanisms of action or pathways involved. Providing more detailed explanations of how zinc, copper, and iron influence skin physiology and pathogenesis would enhance the understanding of their roles in dermatology.
Thank you for your comment, however, we cannot agree with the reviewer on this point. At the beginning of each chapter, we have included a detailed description of the role of individual trace elements not only in human physiology, but also in skin physiology - Zinc - lines: 77-92; Cooper - lines: 218-235; Iron - lines: 329-352. In addition, after the reviewer's suggestion, Figure 1 was modified and also shows the role of these trace elements in skin physiology. In our opinion, the information we have gathered is sufficient to understand the exact role of these trace elements in skin physiology.
The conclusion rightly emphasizes the need for additional research to clarify the role of trace elements in dermatological diseases. However, it does not offer specific suggestions or directions for future studies, such as identifying key research questions, methodologies, etc.
Thank you for the right comment. The summary will be restated as appropriate. After revisions, the summary chapter will refer to the possibility of using ultrasound as one of the research methods that will help assess the exact role of trace elements in skin diseases. In addition, in accordance with the reviewer's suggestion, this manuscript will be given a language correction.
on behalf of the coauthors
Marta Wacewicz-Muczyńska, MD, PhD.
Reviewer 2 Report
Comments and Suggestions for Authors
The authors collected knowledge about the functions and significance of trace element levels of the body in the most frequent skin diseases. They examine the role of zinc, copper, and iron in the possible development and severity of some skin pathologies. The review is well-structured and detailed. They chose an area with limited research. The majority of the literature available is dealing with clinical studies.
All three elements are known as enzyme cofactors, so their low level will cause general enzyme malfunctions and many pathological signs. Iron is regularly measured in laboratories, but the determination of zinc or copper levels is not routine. Also, mainly blood is used, but the element level of RBC or hair is rarely measured. The blood iron level is under complex regulation (inflammation, hemoglobin synthesis demand, Fe/S cluster biogenesis, hypoxia), so it is understandable that the skin manifestation of iron level change is less examined. Also, it is known that the sampling of zinc, for example, has many variables (doi 2023.01.12.23284491). Therefore, the authors had a difficult task of covering the knowledge in this field.
My remarks and questions: Do the authors know how zinc and copper are eliminated from the body? Is there any theory about the effect of high Cu level? The abstract and introduction need some improvement. For example, ln 15: their determination – instead, their level is determined. Some sentences are repeated in the abstract. Ln 39: among other things – please choose another phrase, if possible.
Comments on the Quality of English Language
I wrote some suggestions for it in my comments.
Author Response
Reviewer 2
The authors collected knowledge about the functions and significance of trace element levels of the body in the most frequent skin diseases. They examine the role of zinc, copper, and iron in the possible development and severity of some skin pathologies. The review is well-structured and detailed. They chose an area with limited research. The majority of the literature available is dealing with clinical studies.
Dear Reviewer,
We would like to thank you very much for your careful review of our paper, entitled. " Zinc, copper, and iron in selected skin diseases." and for your accurate and useful suggestions. We have highlighted responses to suggestions in blue italics while changes to the manuscript have been highlighted in green. We hope that the corrections made will prove satisfactory and allow publication of our work in the “International Journal of Molecular Sciences”.
All three elements are known as enzyme cofactors, so their low level will cause general enzyme malfunctions and many pathological signs. Iron is regularly measured in laboratories, but the determination of zinc or copper levels is not routine. Also, mainly blood is used, but the element level of RBC or hair is rarely measured. The blood iron level is under complex regulation (inflammation, hemoglobin synthesis demand, Fe/S cluster biogenesis, hypoxia), so it is understandable that the skin manifestation of iron level change is less examined. Also, it is known that the sampling of zinc, for example, has many variables (doi 2023.01.12.23284491). Therefore, the authors had a difficult task of covering the knowledge in this field.
Thank you for your pertinent commentary, in which the reviewer outlined the most important problems associated with the preparation of the manuscript. The amount of data on changes in trace element concentrations in diseases was very small as a result, the preparation of the manuscript was quite a challenge. We tried to include all source reports related to this issue. We are very flattered by such a positive reviewer's assessment and understanding of the problem involved in the preparation of this paper.
My remarks and questions: Do the authors know how zinc and copper are eliminated from the body?
Thank you for your comment. Zinc is eliminated in small amounts in the urine: daily about 7 mmol/day (0.5 mg/day. This amount increases in patients with liver markosis or diabetes. In addition to urine, zinc is also excreted in feces and sweat (doi: 10.1007/s10238-024-01302-6). As suggested by the reviewer, we have modified the section on zinc physiology accordingly. The changes are highlighted in green Literature added to the article:
- Stiles LI, Ferrao K, Mehta KJ. Role of zinc in health and disease. Clin Exp Med. 2024 Feb 17;24(1):38. doi: 10.1007/s10238-024-01302-6. PMID: 38367035; PMCID: PMC10874324.
- Deep V, Sondhi S, Gupta S. Assessment of Serum Zinc Levels in Patients With Decompensated Cirrhosis of the Liver and Its Association With Disease Severity and Hepatic Encephalopathy: A Prospective Observational Study From North India. Cureus. 2023 Jun 30;15(6):e41207. doi: 10.7759/cureus.41207. PMID: 37525813; PMCID: PMC10387324.
- Marreiro DN, do Perpetuo Socorro C Martins M, de Sousa SS, Ibiapina V, Torres S, Pires LV, do Nascimento Nogueira N, Lima JM, do Monte SJ. Urinary excretion of zinc and metabolic control of patients with diabetes type 2. Biol Trace Elem Res. 2007 Winter;120(1-3):42-50. doi: 10.1007/s12011-007-8000-z. PMID: 17916954.
In the case of copper, the main route of excretion of this element is through the biliary tract. Copper is secreted into the bile, where it then finds its way into the intestines and finally into the feces. Interestingly, the amount of copper that is secreted into the bile is a matter of individual human biology, in addition, some copper is resorbed (doi: 10.3390/ijms21144932). It is estimated that in feces, approx. 0.6-1.6 mg of copper. A small portion of this element is also removed in urine (0.05 mg). We have modified the section on copper accordingly. The changes are marked in green. Added literature:
- Linder Copper Homeostasis in Mammals, with Emphasis on Secretion and Excretion. A Review. Int J Mol Sci. 2020 Jul 13;21(14):4932. doi: 10.3390/ijms21144932. PMID: 32668621; PMCID: PMC7403968.
- Chen J, Jiang Y, Shi H, Peng Y, Fan X, Li C. The molecular mechanisms of copper metabolism and its roles in human diseases. Pflugers Arch. 2020 Oct;472(10):1415-1429. doi: 10.1007/s00424-020-02412-2. Epub 2020 Jun 7. PMID: 32506322.
Is there any theory about the effect of high Cu level?
It is unfortunate that there are currently no data on high copper concentrations in dermatological patients. We are not in a position to say that high copper concentrations are the cause of the disease or rather its effect. It is possible that in the near future such data will emerge from the scientific community.
The abstract and introduction need some improvement. For example, ln 15: their determination – instead, their level is determined. Some sentences are repeated in the abstract. Ln 39: among other things – please choose another phrase, if possible.
Thank you for your comment and we apologize for the confusion. We have modified the introduction accordingly and included all the reviewer's suggestions. All changes have been highlighted in green.
In conclusion, we would like to thank you once again for your thorough and accurate analysis of our manuscript and your accurate comments, which will certainly enrich our work. We hope that the corrections made will satisfy the reviewer and the paper can be published in the International Journal of Molecular Sciences. In addition, in accordance with the suggestion of the first reviewer, this manuscript will be given a language correction.
Marta Wacewicz-Muczynska, MD, PhD.
Also on behalf of the co-authors
Reviewer 3 Report
Comments and Suggestions for Authors
This review presents research’s aimed at elucidating the roles of zinc, copper, and iron in skin physiology, while also compiling and analyzing data pertaining to these elements in various skin diseases. It highlights their potential utility as adjunctive diagnostic tools in dermatology.
No plagiarism was detected. This review is interesting and will attract the attention of the people working in the field of dermatology, bioinorganic chemistry medicinal chemistry.
However, the novelty of this study concerning the existing recent review on the “Trace element zinc and skin disorders” has been published by Zou et.al., (2023) Front. Med. 9:1093868. doi: 10.3389/fmed.2022.109386 must be validated before this article can be accepted for publication.
Moreover, the review on “Iron, Copper, and Zinc Concentrations in Normal Skin and in Various Nonmalignant and Malignant Lesions” by Corodetsky et.al in INTERNATIONAL JOURNAL OF DERMATOLOGY, 1986, Vol. 25, 440; as well as the report on Determination of Zinc, Copper, Manganese, and Iron in Blood from Patients with Light-Sensitive Skin Diseases by Horkaya, in Dcrmatologica 169: 66-69 (1984) are missing from the references.
Abstract
Skin diseases that are investigated concerning the presence or absence of zinc, copper, and iron should be reported in the Abstract
Introduction
“ if the intake of trace elements exceeds the so-called critical level, they begin to have a toxic effect “ The text implies that all trace elements, when consumed in excess, have a toxic effect on the body and contribute to disease development. While this is true for some trace elements, it's not universally applicable to all.
2.1 Biological Roles of Zinc in the Human Body
"NK cells" (Natural Killer cells) abbreviations should be defines once they appear first.
2.2. Zinc in skin physiology
It is essential to provide the definition of the unit "μg/g."
2.3. Zinc in Psoriasis and Pemphigus vulgaris
"The condition significantly diminishes patients' quality of life and, coupled with an elevated risk of systemic diseases, contributes to premature mortality [32-34]" the systemic diseases which are associated with psoriasis, should be specifying.
Comments on the Quality of English LanguageThis review presents research’s aimed at elucidating the roles of zinc, copper, and iron in skin physiology, while also compiling and analyzing data pertaining to these elements in various skin diseases. It highlights their potential utility as adjunctive diagnostic tools in dermatology.
No plagiarism was detected. This review is interesting and will attract the attention of the people working in the field of dermatology, bioinorganic chemistry medicinal chemistry.
However, the novelty of this study concerning the existing recent review on the “Trace element zinc and skin disorders” has been published by Zou et.al., (2023) Front. Med. 9:1093868. doi: 10.3389/fmed.2022.109386 must be validated before this article can be accepted for publication.
Moreover, the review on “Iron, Copper, and Zinc Concentrations in Normal Skin and in Various Nonmalignant and Malignant Lesions” by Corodetsky et.al in INTERNATIONAL JOURNAL OF DERMATOLOGY, 1986, Vol. 25, 440; as well as the report on Determination of Zinc, Copper, Manganese, and Iron in Blood from Patients with Light-Sensitive Skin Diseases by Horkaya, in Dcrmatologica 169: 66-69 (1984) are missing from the references.
Abstract
Skin diseases that are investigated concerning the presence or absence of zinc, copper, and iron should be reported in the Abstract
Introduction
“ if the intake of trace elements exceeds the so-called critical level, they begin to have a toxic effect “ The text implies that all trace elements, when consumed in excess, have a toxic effect on the body and contribute to disease development. While this is true for some trace elements, it's not universally applicable to all.
2.1 Biological Roles of Zinc in the Human Body
"NK cells" (Natural Killer cells) abbreviations should be defines once they appear first.
2.2. Zinc in skin physiology
It is essential to provide the definition of the unit "μg/g."
2.3. Zinc in Psoriasis and Pemphigus vulgaris
"The condition significantly diminishes patients' quality of life and, coupled with an elevated risk of systemic diseases, contributes to premature mortality [32-34]" the systemic diseases which are associated with psoriasis, should be specifying.
Author Response
Reviewer 3
This review presents research’s aimed at elucidating the roles of zinc, copper, and iron in skin physiology, while also compiling and analyzing data pertaining to these elements in various skin diseases. It highlights their potential utility as adjunctive diagnostic tools in dermatology.
Dear Reviewer,
We would like to thank you very much for your careful review of our paper, entitled. " Zinc, copper, and iron in selected skin diseases." and for your accurate and useful suggestions. We have highlighted responses to suggestions in blue italics while changes to the manuscript have been highlighted in green. We hope that the corrections made will prove satisfactory and allow publication of our work in the “International Journal of Molecular Sciences”.
No plagiarism was detected. This review is interesting and will attract the attention of the people working in the field of dermatology, bioinorganic chemistry medicinal chemistry.
Thank you for such a positive comment regarding our manuscript.
However, the novelty of this study concerning the existing recent review on the “Trace element zinc and skin disorders” has been published by Zou et.al., (2023) Front. Med. 9:1093868. doi: 10.3389/fmed.2022.109386 must be validated before this article can be accepted for publication.
Thank you for your comment. We would like to refer to the paper mentioned by the reviewer - Zou P, Du Y, Yang C, Cao Y. Trace element zinc and skin disorders. Front Med (Lausanne). 2023 Jan 17;9:1093868. doi: 10.3389/fmed.2022.1093868. PMID: 36733937; PMCID: PMC9887131. The author, like our team, has published a review paper on the role of zinc in dermatological diseases. However, it should be noted that the amount of similarity between the paper by Zou et al. (2023) and our manuscript is low. Like our paper, Zou et al. (2023) focuses on the physiological role of zinc; however, unlike us, it does not describe the exact role of this trace element in skin physiology. Another important difference is that Zou et al. (2023) describes a much larger number of conditions than we do in our manuscript. This is due to the fact that the authors focused exclusively on the role of zinc, while in our manuscript copper and iron are additionally described. Therefore, we focused on a smaller number of disease entities, selecting the conditions most common in the population for the paper. In addition, Zou et al. (2023) describes not only skin diseases, but also metabolic disorders (e.g. Acrodermatitis enteropathica; Necrolytic migratory erythhema), hair diseases (e.g. Alopecias) and also cancer. We would also like to, emphasize that we feel that the description of the role of zinc in our selected diseases (Psoriasis, Pemphigus vulgaris, Atopic Dermatitis and Seborrheic Dermatitis), however, is much more elaborate than in the paper by Zou et al. (2023). In addition, our manuscript is not limited to studies conducted on serum, but also presents studies on hair.
In conclusion, there are some similarities between the work of Zou et al. (2023) and the present manuscript such as the selection of diseases described and the type of study material, however, the two papers differ significantly. The most important difference comes from the fact that our manuscript describes copper and iron in addition to zinc.
Moreover, the review on “Iron, Copper, and Zinc Concentrations in Normal Skin and in Various Nonmalignant and Malignant Lesions” by Corodetsky et.al in INTERNATIONAL JOURNAL OF DERMATOLOGY, 1986, Vol. 25, 440; as well as the report on Determination of Zinc, Copper, Manganese, and Iron in Blood from Patients with Light-Sensitive Skin Diseases by Horkaya, in Dcrmatologica 169: 66-69 (1984) are missing from the references.
Thank you very much for your comment. We would like to point out that we analyzed the work “Gorodetsky R, Sheskin J, Weinreb A. Iron, copper, and zinc concentrations in normal skin and in various nonmalignant and malignant lesions. Int J Dermatol. 1986 Sep;25(7):440-5. doi: 10.1111/j.1365-4362.1986.tb03449.x. PMID: 3771040.” during the preparation of this manuscript. However, due to the selection of the study group - patients with solar keratosis, chronic solar dermatitis and skin cancers (squamous cell carcinoma and malignant melanoma) we decided not to include this work in our manuscript. In addition, trace element determinations were performed using a fluorescence method that did not match the methods selected in our manuscript. Therefore, we will not include this work in our manuscript.
For the paper "Horkay I, Tehrani DK, Altmann H, Krajczár J. Determination of zinc, copper, manganese, and iron in blood from patients with light-sensitive skin diseases. Dermatologica. 1984;169(2):66-9. doi: 10.1159/000249570. PMID: 6479415, we are dealing with diseases that were not included in our manuscript - polymorphic light eruption and patients with cutaneous porphyrias. Although the trace elements in this study were determined from blood, we will not be adding this work to the manuscript due to the fact that the disease entities described by Horkay et al. (1984) do not match the disease entities selected for our manuscript.
Abstract
Skin diseases that are investigated concerning the presence or absence of zinc, copper, and iron should be reported in the Abstract
Thank you very much for your pertinent comment. We agree with the reviewer's suggestion that we wrongly failed to include the diseases we described in the abstract. This is a big mistake, for which we apologize. We have modified the abstract accordingly - in accordance with the reviewer's suggestion. We hope that after the corrections it will be better in perception and will be satisfactory to the reviewer. All changes have been marked in green.
Introduction
“ if the intake of trace elements exceeds the so-called critical level, they begin to have a toxic effect “ The text implies that all trace elements, when consumed in excess, have a toxic effect on the body and contribute to disease development. While this is true for some trace elements, it's not universally applicable to all.
Thank you for your accurate comment. We agree with the reviewer that only certain trace elements consumed in excess have harmful effects on the body. We have modified the given passage accordingly so as not to mislead the potential reader. At the same time, we would like to apologize very much, for the inaccuracy in the given passage.
2.1 Biological Roles of Zinc in the Human Body
"NK cells" (Natural Killer cells) abbreviations should be defines once they appear first.
Thank you for your comment and we apologize for the mistake. We have corrected the error in question. The correction has been highlighted in green.
2.2. Zinc in skin physiology
It is essential to provide the definition of the unit "μg/g."
Thank you very much for your rightful comment. The definition of the unit has been introduced into the manuscript, as suggested by the reviewer. The change has been highlighted in green.
2.3. Zinc in Psoriasis and Pemphigus vulgaris
"The condition significantly diminishes patients' quality of life and, coupled with an elevated risk of systemic diseases, contributes to premature mortality [32-34]" the systemic diseases which are associated with psoriasis, should be specifying.
Thank you for your accurate comment. In accordance with the reviewer's suggestion, we have specified in the manuscript which systemic diseases are associated with psoriasis. The given passage has been modified accordingly, according to the reviewer's suggestions and any changes have been marked in green.
Thank you for the right comment. The summary will be restated as appropriate. After revisions, the summary chapter will refer to the possibility of using ultrasound as one of the research methods that will help assess the exact role of trace elements in skin diseases. In addition, in accordance with the reviewer's suggestion, this manuscript will be given a language correction.
on behalf of the coauthors
Marta Wacewicz-Muczyńska, MD, PhD.
Round 2
Reviewer 1 Report
Comments and Suggestions for Authors
The authors provided a comprehensive response to the reviewer's comment, considering the suggestions/criticisms to improve the article.
Consequently, I recommend the publication.
Comments on the Quality of English LanguageMinor editing of English language required
Author Response
Reviewer 1
The authors provided a comprehensive response to the reviewer's comment, considering the suggestions/criticisms to improve the article.
Consequently, I recommend the publication.
Thank you very much for such positive feedback on our manuscript. We are glad that the changes made were accepted by the reviewer, we would like to emphasize that all the suggestions suggested by the reviewer were very helpful and enriched our manuscript.
Marta Wacewicz-Muczynska, MD, PhD.
Also on behalf of the co-authors
Reviewer 3 Report
Comments and Suggestions for Authors
Almost all of the comments made on the original version of the article have been answered by the authors or taken into account in the preparation of the revised version, but there remains one point that needs to be clarified before the article can be accepted for publication.
What does '(60 μg/g [micrograms per gram])' refer to? is it shows 60 μg of zinc in g of what. Please provide a definition."
Comments on the Quality of English Languagethe English used correct and readable
Author Response
Reviewer 3
Almost all of the comments made on the original version of the article have been answered by the authors or taken into account in the preparation of the revised version, but there remains one point that needs to be clarified before the article can be accepted for publication.
Thank you very much for such positive feedback and insightful review of our manuscript. The changes suggested by the reviewer have greatly enriched the content of the manuscript.
What does '(60 μg/g [micrograms per gram])' refer to? is it shows 60 μg of zinc in g of what. Please provide a definition."
Thank you for your comment and we apologize for the inaccuracy. The unit described refers to the amount of μg of zinc per gram of skin tissue (epidermis and dermis, respectively). We have corrected the passage in question accordingly, changes highlighted in green.
In conclusion, we would like to thank you once again for your thorough and accurate analysis of our manuscript and your accurate comments, which will certainly enrich our work. We hope that the corrections made will satisfy the reviewer and the paper can be published in the International Journal of Molecular Sciences.
Marta Wacewicz-Muczynska, MD, PhD.
Also on behalf of the co-authors

Round 3
Reviewer 3 Report
Comments and Suggestions for Authors
The manuscript can now be published in IJMS.